

# Relationship between emotional intelligence and empathy towards humans and animals

Raquel Gómez-Leal[1], Ana Costa[2], Alberto Megías-Robles[1],
Pablo Fernández-Berrocal[1] and Luísa Faria[2]

[1] Department of Basic Psychology, Faculty of Psychology, University of Málaga, Málaga, Spain
[2] Faculty of Psychology and Education Sciences, University of Porto, Porto, Portugal

## ABSTRACT

Previous research has highlighted that Emotional Intelligence (EI) is related to an array of positive interpersonal behaviours, including greater human empathy. Nonetheless, although animals are an integral part of our lives, there is still a lack of clarity regarding the way in which EI relates to empathy towards animals. The aim of this study was to investigate the relationship between EI and empathy towards humans and animals. We used the Trait-Meta Mood Scale to assess EI, the Interpersonal Reactivity Index to assess empathy for humans, and the Animal Empathy Scale to assess empathy for animals. Our findings revealed a positive relationship between empathy for humans and animals. The results also supported the idea that EI is positively related to empathy for humans, while the relationship between EI and empathy for animals was dependent on whether or not the participants had experience with pets. In addition, multiple regression analysis showed that the variables that best predicted empathy for animals were having a pet (or not), age, gender and human empathic concern. Finally, the relationship between human empathic concern and empathy for animals was stronger in participants who had pets. These findings provide a better understanding of the mechanisms underlying empathic behaviour and suggest that empathy for humans and animals can be influenced by different factors. Limitations and future lines of research are discussed.

## INTRODUCTION

Nowadays it is generally accepted that the intelligent use of emotions has a positive impact on the psychological adaptation of the individual to their environment (*Mayer & Salovey, 1997*; *Salovey et al., 1999*; *Salovey et al., 1995*), providing them with a better chance of success (*Mayer, Roberts & Barsade, 2008*). Based on this perspective, research conducted within the field of emotional intelligence (EI) has made a significant contribution to knowledge and evidence regarding the positive effects of emotions. In particular, research in recent decades indicates that an array of positive outcomes can be attributed to higher levels of EI, including improved well-being and mental health (*Martins, Ramalho & Morin, 2010*; *İnce, Şimsek & Özbek, 2019*), academic or professional performance (*Costa &*

Corresponding author
Raquel Gómez-Leal,
raqgomlea@uma.es

*Faria, 2015*; *O'Boyle et al., 2011*), prosocial behaviour and satisfaction with social networks (*Ciarrochi, Chan & Caputi, 2000*; *Mayer, Caruso & Salovey, 1999*), lower levels of clinical symptomatology (*Bastian, Burns & Nettelbeck, 2005*; *Megías et al., 2018a*) and aggressive or disruptive behaviour (*Brackett, Mayer & Warner, 2004*; *Davis & Humphrey, 2012*; *Lopes et al., 2011*; *Megías et al., 2018b*).

In this regard, research has also been devoted towards exploring the relationship between EI and empathetic behaviours, namely the positive effects of EI on empathy for other humans. Nonetheless, to date no research has addressed the specific relationship between EI and empathy for animals, in spite of the fact that animals play a very important role in our society, and are an integral part of culture, leisure, well-being, work, and politics. In fact, public opinion would suggest that people who show sensitivity to nonhuman species have greater emotional abilities. However, the analysis of well-known cases, such as activists who violate human rights to save animals or even Hitler and his Nazi companions who were animal lovers (*Paton, 1993*), demonstrate that empathy for animals may not always relate to empathy for humans. This study presents a preliminary attempt to extend knowledge on the relationship between EI and empathy towards humans and animals.

## EI and empathy towards humans

EI can be conceptualized as the capacity to process emotional information and comprises the "ability to perceive accurately, appraise, and express emotion; the ability to access and/ or generate feelings when they facilitate thought; the ability to understand emotion and emotional knowledge; and the ability to regulate emotions to promote emotional and intellectual growth" (*Mayer & Salovey, 1997*, p. 10). Thus, both intrapersonal and interpersonal emotional abilities are considered to fall under this category of mental abilities (*Mayer & Salovey, 1997*).

Particular interest has been paid to the link between EI and empathy, since the latter constitutes a relevant factor in social interaction and prosocial behaviour (*Gilet et al., 2013*). Empathy, as a multidimensional construct that comprises emotional, cognitive and motivational components (*Baron-Cohen & Wheelwright, 2004*; *Cuff et al., 2014*), is based on the abilities to recognize, understand, and share the feelings of others (*Davis, 1980*; *De Waal, 2008*; *Preston & De Wall, 2002*). More specifically, cognitive empathy reflects the way in which we understand others, their experiences and emotions, emotional empathy involves the emotional response to the experience of others and actually sharing that particular emotional state with the other (*Smith, 2006*), which often generates an empathic concern, understood as compassion or motivational empathy, which leads a person to take action to relieve the suffering of others (*Eisenberg & Miller, 1987*; *Pfattheicher, Sassenrath & Schindler, 2015*).

Considering that perceiving and understanding emotion in others and emotional awareness are abilities involved in EI, it might be reasonable to suppose that there is a positive relationship between EI and empathy (*Schutte et al., 2001*). In fact, there are parallels between some of the features of EI and empathy. *Petrides, Furnham & Martin (2004)* found evidence to suggest that the trait EI models comprise affect-related

functioning such as emotional awareness, empathy and relationship skills. According to *Mayer & Salovey (1997)*, an individual with optimum EI can better perceive, understand, and manage their own emotions, and are more likely to be skilled at generalizing these abilities of perceiving, understanding, and managing to the emotions of others. Some authors have even argued that empathy is a result of EI, since the ability to reason about emotions in ourselves and others will have an impact on the accurate interpretation and management of social interactions and emotional experiences (*Mayer, Roberts & Barsade, 2008*).

Various authors have delved further into this relationship and confirmed that individuals with higher EI are also more empathetic towards other people (*Fitness & Curtis, 2005*; *Mayer, Caruso & Salovey, 1999*; *Schutte et al., 2001*). This positive relationship has been established when evaluating EI using different types of measures, including self-report (*Fitness & Curtis, 2005*; *Salovey et al., 2002*; *Schutte et al., 2001*) and performance tests (*Ciarrochi, Chan & Caputi, 2000*; *Mayer, Caruso & Salovey, 1999*). In particular, some studies found that attention to emotions correlated positively with the empathy dimensions of empathic involvement and personal distress (*Aguilar-Luzón & Augusto, 2009*; *Extremera & Fernández-Berrocal, 2004*). A higher level of emotional clarity and repair has also been positively associated with perspective taking and negatively associated with personal distress, both of which are aspects of empathic behaviour (*Aguilar-Luzón & Augusto, 2009*; *Extremera & Fernández-Berrocal, 2004*; *Ramos, Fernandez-Berrocal & Extremera, 2007*).

Although the relationship between EI and human empathy has been explored in the literature, rather less attention has been paid to the issue of how EI relates to empathy directed towards other entities, including empathy for animals. Given that empathy is related to the socio-emotional abilities to recognize, understand and share the feelings of others, and that understanding and being aware of the emotional signs in animals (with more or less phylogenetic proximity to humans) can pose particular challenges since this is quite distinct from human interactions, the relationship between EI and empathy towards animals could be quite different from the one established with humans.

## Relationship between empathy directed to humans and to animals

Over the past few decades, public opinion has shifted from the traditional conceptions of animals as objects to be used by humans to a broader ethical perspective of care and compassion towards them. In fact, the public attitudes to animals related to increasing sensitivity and concern about animal use have developed in parallel with the stronger beliefs about the ability of animals to experience pain and suffering, along with their cognitive abilities and their sentience (*Cornish et al., 2018*). This progressive change in society's attitudes towards animals is most likely to be based on the increasing proximity with animals in our daily life (e.g., pets) and on the countless contexts in which housed animals such as zoos, aquariums, museums, sanctuaries, shelters, nature centres and others offer opportunities to have educational experiences with animals and nature (*Young, Khalil & Wharton, 2018*).

Within the broader research area on the nature of empathy, the earliest studies exploring the relationship between human individuals and other animals emerged and, in particular, these works demonstrated that humans are able to feel empathy for animals (e.g., *Emauz et al., 2016*; *Paul, 2000*). Moreover, empathy towards animals seem to have originated in a similar way as that shown towards other humans (*Ascione, 1992*; *Kohl, 2012*; *Ruckert, 2016*). In his precursory research, *Paul (2000)* elaborated on the previous empathy definition put forward by *Eisenberg (1995)* and specified that empathy towards animals entailed a vicarious emotional response to the emotions or states of animals, and the cognitive understanding of their thoughts or feelings. For *Drane (2009)* empathy is the ability to feel what others are feeling, regardless of whether this comes from a direct relationship between humans or animals. Jorge Ritchman (cited in *García, 2014*) also extrapolated empathy towards our relationship with animals, considering this to be fundamental to our coexistence, since it allows us to perceive the damage that we can cause to other species, feel their suffering, or avoid it (*García, 2014*). More recently, a metanalytic review on empathy for animals defined empathy as a simulated emotional state that relies on the ability to perceive, understand and care about the experiences or perspectives of another person or animal (*Young, Khalil & Wharton, 2018*). Therefore, empathy towards animals comprises the same three abilities as empathy towards humans–affective empathy, cognitive empathy, and empathic concern (*Cuff et al., 2014*). Affective empathy is the ability to sense or physically experience the emotions of another (*Cuff et al., 2014*). For instance, when an individual observes an animal in a state of suffering, he/she may experience distress as if they were responding to the same stimulus (*Eres et al., 2015*). Cognitive empathy is the ability to understand the experiences of others by recognizing and imagining their reality (*Cuff et al., 2014*), and might support (or not) our affective empathy. For instance, it supports our affective empathy when we believe the animal is suffering because we recognized that it is physically injured; or it does not provide such support, when, for instance, we understand that an animal is isolated due to their specific biological characteristics and it is not a sign of depression. Empathic concern, on the other hand, can motivate a person to take action and relieve the suffering of the animal (*Eisenberg & Miller, 1987*; *Pfattheicher, Sassenrath & Schindler, 2015*), and in that case, an individual would help an animal that is injured or trapped.

Some authors consider that empathy for animals has a strong heritable component and can evolve differently depending on the particular species of animals (*Bradshaw & Paul, 2010*). Research suggests that the development of empathic behaviour is due to its adaptive components, which would enable pro-social behaviour and inhibit aggression. Another possibility explored by some investigators is that the process of nurturing (e.g., providing food and shelter, care-giving) infants and babies would have had an impact on the development of the empathic behaviours of humans, considering that the ability to empathetically respond to the distress shown by children is a crucial component of the emotional nurturance process (*De Waal, 2008*).

Moreover, the literature also indicates that there is a positive relationship between the empathy directed to humans and animals, although this is not of a high magnitude (*Ellingsen et al., 2010*; *Emauz et al., 2016*; *Paul, 2000*). Other studies have also found that

concerns about animal suffering are associated with higher levels of empathy for humans (*Ascione, 1992*; *Komorosky & O'Neal, 2015*). However, when exploring whether individuals particularly characterized by high levels of affection towards animals have high levels of affection towards humans, the results are contradictory (*Paul, 2005*). For instance, a very high level of affection for animals can be related to a displacement of affection from people to pets (*Veevers, 1985*). Therefore, it is not always evident that in order to be empathetic towards animals the individual should also be empathetic towards humans or vice versa. These findings suggest that empathy for humans and for animals—whilst many times related—are probably not the same unitary construct, representing different psychological concepts or, at least, separately influenced by specific factors (*Paul, 2000*; *Paul, 2005*). Perhaps the possibility that both types of empathy have shared and non-shared components or because they act under the influence of specific moderator mechanisms could explain why differences are often observed in the empathic responses shown towards human individuals and other animals (*Paul, 2000*).

## EI and empathy towards animals

To date, there are no reports regarding the relationship between EI and empathy directed to animals of any kind, except humans. One could argue that EI is related to several positive emotional outcomes such as empathy for humans (*Fitness & Curtis, 2005*; *Mayer, Caruso & Salovey, 1999*; *Schutte et al., 2001*) and that a similar association is likely to be found for empathy towards animals. However, as previously described, the literature on empathy for animals has presented mixed results, that is, supporting the association between attitudinal and prosocial behaviours towards animals and towards people (*Ellingsen et al., 2010*; *Emauz et al., 2016*; *Komorosky & O'Neal, 2015*) or indicating that those behaviours can be independent (*Paul, 2005*). Moreover, the dynamics of the relationships between individuals are far more complex than those established with animals. One possibility is that the ability to understand and manage emotions based on human interactions might prove to be insufficient to perceive and understand the emotions of animals, which hinders the capacity of humans to empathise with them. Therefore, it is possible that for there to be an association between EI and empathy for animals, the influence of other variables—such as the person's current experience with animals—could be critical. In this regard, EI can be associated with empathy for humans and other positive human-oriented outcomes but might not necessarily be correlated with animal-oriented constructs, for instance, advocating for animal rights, being compassionate towards animals in distress, or taking a stance against the use of animals for scientific purposes.

## Gender differences in EI and empathy

Previous studies in the literature have revealed gender differences in EI, and in empathy for humans and animals. Whilst in general, women score higher than men on the main factors that constitute EI, this difference appears to depend on the type of instrument used. Specifically, when a performance-based instrument is used, women score higher than men on all dimensions; however, when using a self-report instrument, particularly the TMMS

(*Salovey et al., 1995*), women tend to score higher than men on the dimension of attention to emotions, and lower on the dimensions of emotional clarity and repair (*Cabello & Fernández-Berrocal, 2015*; *Fernández-Berrocal & Extremera, 2008*; *Joseph & Newman, 2010*; *Navarro-Bravo et al., 2019*). Further, previous studies indicate that women, compared with men, tend to exhibit higher levels of empathy for both humans and animals (*Angantyr, Eklund & Hansen, 2011*; *Klein & Hodges, 2001*; *Paul, 2000*; *Serpell, 2004*).

### Aim

The main objective of this research was to investigate the relationship between EI and empathy for humans and animals. We conducted a detailed study through the analysis of several EI and empathy sub-dimensions. In addition, we examined the possible effect of previous experience with animals on these relationships. Based on the findings of the previous literature, we also explored possible gender-related differences in the scores of EI and empathy. We hypothesized that (1) there is a positive relationship between EI and empathy for animals; (2) there is a positive relationship between empathy for humans and empathy for animals; (3) with respect to the relationship between EI and empathy for animals, we conducted an exploratory analysis since it is not possible to formulate a clear hypothesis given the mixed findings reported in the previous literature; and finally, (4) we proposed that the relationships between EI, human empathy and empathy for animals may depend on the degree of proximity between the individual and animals (operationalized according to whether they have pets or not).

## METHODS

### Participants

The sample was composed of four hundred and seventy-one adult volunteers (34.4% male). They were recruited through advertisements at the University of Málaga, social networks, and online platforms. The age of the participants ranged from 18 to 65 years with a mean of 26.15 years (SD = 10.10). Two hundred and fifty-eight of the participants were pet owners. All participants were informed that confidentiality and anonymity of the collected data would be assured, and they were treated in accordance with the Helsinki declaration (*World Medical Association, 2008*). The study was approved by The Research Ethics Committee of the University of Málaga as part of the project PSI2017-84170-R (IRB approval number CEUMA 14-2019-H).

### Procedure and instruments

Online questionnaires were completed by the participants through the LimeSurvey platform (http://limesurvey.org). The respondents accessed the questionnaires via an email link sent by the authors. An informed consent form was included in the survey, and the participants were assured of confidentiality and anonymity. To avoid missing data, the questionnaires were set up so that blank responses were not allowed. For each participant, this entire process took approximately 20 min to complete.

A description of each scale is detailed below:

Trait-Meta Mood Scale (TMMS; *Salovey et al., 1995*). The TMMS is a 24-item self-report scale widely used to assess EI. The questionnaire includes three sub-dimension scores: attention to emotions (awareness of our emotions, the ability to recognize our feelings and know what they mean), emotional clarity (ability to know, understand, distinguish and understand how emotions evolve, ability to integrate emotions in our thinking), and emotional repair (ability to regulate and control positive and negative emotions). Responses are given on a 5-point Likert type scale ranging from 1 ("Disagree strongly") to 5 ("Agree strongly"). We used the Spanish version of the scale (*Fernández-Berrocal, Extremera & Ramos, 2004*). In our study, the scale showed good internal consistency (Cronbach's alpha values of the sub-dimensions ranged between 0.85 and 0.91).

Interpersonal Reactivity Index (IRI; *Davis, 1983*) is a 28-item self-report scale used to measure empathy. This scale is composed of four sub-dimensions: perspective taking (ability of subjects to adopt other people's point of view), empathic concern (tendency of subjects to experience feelings of compassion and concern towards others), personal distress (tendency of subjects to experience feelings of anxiety and discomfort when witnessing the negative experiences of others) and fantasy (tendency of subjects to identify with fictional characters from books and movies). Each item uses a 5-point Likert scale ranging from 1 ("Does not describe me at all") to 5 ("Describes me very well"). We used the Spanish version of the scale (*Escrivá, Navarro & García, 2004*). In our study, the scale showed adequate internal consistency (Cronbach's alpha value of the total score was 0.79 and for the sub-dimensions this ranged between 0.66 and 0.76).

Animal Empathy Scale (AES; *Paul, 2000*) is a 22-item self-report scale used to measure empathy for animals through the assessment of the individual's feelings about animals and their treatment. The scale comprises items that enquire about both empathic relationships (e.g., "It makes me sad to see an animal on its own in a cage"; "It upsets me when I see helpless old animals") and non-empathic relationships (e.g., "Dogs sometimes whine and whimper for no real reason"; "Sometimes I am amazed how upset people get when an old pets dies"). Responses are scored by a 9-point Likert scale ranging from 1 ("Disagree strongly") to 9 ("Agree strongly"). We used the Spanish version of the scale in our study (*La Torre Gómez, 2017*). In our study, the scale showed adequate internal consistency (Cronbach's alpha value of the total score was 0.87).

## Data analysis

First, descriptive statistics were computed to examine the characteristics of the scores of the measures employed, both for the total sample and divided by gender. Second, gender differences were contrasted using *t*-tests. Third, differences according to pet and non-pet ownership were examined by *t*-tests. Fourth, Pearson's correlations were conducted to explore associations between the study variables. Fifth, in order to verify if the association between empathy for animals and the variables of empathy for humans and EI are influenced by having pets, additional Pearson's correlation analyses were carried out by dividing the sample into pet owners and non-pet owners. Besides, using Fisher's *Z*-test,

**Table 1 Means, standard deviations (SD), and *t*-test for gender differences.**

| | Total sample | | Men | | Women | | | |
|---|---|---|---|---|---|---|---|---|
| | Mean | SD | Mean | SD | Mean | SD | *t*-test | Cohen's *d* |
| Attention (TMMS) | 3.35 | 0.88 | 3.15 | 0.89 | 3.45 | 0.86 | −3.49** | 0.34 |
| Clarity (TMMS) | 3.16 | 0.83 | 3.24 | 0.83 | 3.12 | 0.84 | 1.59 | 0.14 |
| Repair (TMMS) | 3.17 | 0.76 | 3.23 | 0.74 | 3.15 | 0.77 | 1.11 | 0.11 |
| Perspective-taking (IRI) | 3.57 | 0.67 | 3.46 | 0.68 | 3.63 | 0.66 | −1.21** | 0.25 |
| Fantasy (IRI) | 3.23 | 0.70 | 3.06 | 0.66 | 3.32 | 0.70 | −4.01** | 0.40 |
| Empathic concern (IRI) | 4.00 | 0.61 | 3.66 | 0.67 | 4.16 | 0.50 | −9.19** | 0.85 |
| Personal distress (IRI) | 2.40 | 0.77 | 2.25 | 0.75 | 2.49 | 0.77 | −3.19** | 0.32 |
| Animal empathy (AES) | 148.39 | 28.28 | 139.84 | 27.03 | 152.85 | 27.97 | −4.85** | 0.47 |

Note:
** $p < 0.01$.

we tested if there were significant differences between the human empathy and EI correlation and the animal empathy and EI correlation, in the latter case for both the total sample and the sample divided according to pet ownership status. Sixth, we conducted hierarchical multiple regression analyses in order to identify the study variables most strongly associated with empathy for animals. Pet ownership, gender and age were entered at Step 1, the three EI sub-dimensions were entered at Step 2, and the four human empathy sub-dimensions were entered at Step 3. Finally, we explored the possible moderating effect of pet ownership on the significant relationships observed in the previous regression analysis. Descriptive statistics, *t*-tests, Pearson's correlations, Fisher's Z-test and regression analyses were carried out using SPSS® version 24.0 (IBM Corporation, Armonk. NY, USA) and FZT computator (http://psych.unl.edu/psycrs/statpage/regression.html). SPSS PROCESS macro 2.16, Model 1 (*Hayes, 2013*) was used for the moderation analysis.

## RESULTS

Table 1 displays the descriptive statistics and gender differences for the variables included in the study. We observed that women, in comparison with men, scored higher on the attention to emotions sub-dimension of EI ($p < 0.01$), in the perspective-taking, empathic concern, fantasy, and personal distress sub-dimensions of human empathy ($p < 0.01$) and on the scale of empathy for animals ($p < 0.01$). *T*-tests comparing pet owners and non-pet owners only revealed significant differences on the scale of empathy for animals, where pet owners obtained a higher score ($p < 0.01$; see Table 1 for more details).

Pearson's correlation analysis including the total sample (see Table 2) confirmed that scores on the attention to emotions sub-dimension of EI were positively related to all sub-dimensions of human empathy ($ps < 0.05$); the emotional clarity sub-dimension was negatively related to personal distress, and positively related to fantasy and perspective-taking ($ps < 0.05$); and the emotional repair sub-dimension was positively related to perspective-taking and fantasy, and negatively related to personal distress ($ps < 0.05$). Regarding empathy for animals, the results revealed a positive relationship between the levels of empathy for animals and human empathy for the sub-dimensions of

**Table 2 Pearson's correlations between EI and empathy for humans in the total sample.**

|  | 2 | 3 | 4 | 5 | 6 | 7 | 8 |
|---|---|---|---|---|---|---|---|
| 1. Attention (TMMS) | 0.27** | 0.06 | 0.32** | 0.25** | 0.35** | 0.20** | 0.13** |
| 2. Clarity (TMMS) | – | 0.35** | 0.12* | 0.10* | −0.02 | −0.26** | −0.04 |
| 3. Repair (TMMS) |  | – | 0.13** | 0.24** | 0.08 | −0.24** | 0.07 |
| 4. Perspective-taking (IRI) |  |  | – | 0.25** | 0.33** | −0.11* | 0.13** |
| 5. Fantasy (IRI) |  |  |  | – | 0.35** | 0.22** | 0.20** |
| 6. Empathic concern (IRI) |  |  |  |  | – | 0.16** | 0.29** |
| 7. Personal distress (IRI) |  |  |  |  |  | – | 0.03 |
| 8. Animal empathy (AES) |  |  |  |  |  |  | – |

Notes:
\* $p < 0.05$.
\*\* $p < 0.01$.

**Table 3 Pearson's correlations between EI and empathy for humans and animals according to pet ownership.**

|  | Attention (TMMS) | Clarity (TMMS) | Repair (TMMS) | Perspective-taking (IRI) | Fantasy (IRI) | Empathic concern (IRI) | Personal distress (IRI) |
|---|---|---|---|---|---|---|---|
| Animal empathy (AES) (Pet owners) | 0.18** | −0.02 | 0.14* | 0.13* | 0.27** | 0.40** | 0.05 |
| Animal empathy (AES) (Non-pet owners) | 0.11 | −0.04 | 0.02 | 0.14* | 0.12 | 0.15* | 0.04 |

Notes:
\* $p < 0.05$.
\*\* $p < 0.01$.

perspective-taking, fantasy and empathic concern. Moreover, higher levels of empathy for animals were related to higher EI, but only for the sub-dimension of attention to emotions.

Pearson's correlation analyses conducted by dividing the sample into pet owners and non-pet owners (see Table 3) revealed a positive relationship between empathy for animals and human empathy for the sub-dimensions of perspective-taking and empathic concern in both samples. Moreover, a positive relationship was also revealed between empathy for animals and the human empathy sub-dimension of fantasy, but only for the sample of pet owners. With respect to the relationship between empathy for animals and EI, in the sample of non-pet owners, empathy for animals was not related to any of the sub-dimensions of EI. However, in the sample of pet owners, the results showed that higher levels of empathy for animals were related to higher EI for the sub-dimensions of attention to emotions and repair.

When specific comparisons between correlations were made using Fisher's $z$-test, it was found that the EI sub-dimension of attention to emotions showed a significantly stronger positive relationship with the human empathy sub-dimensions of perspective-taking ($Z = 3.07$, $p < 0.01$) and empathic concern ($Z = 3.59$, $p < 0.01$) than with empathy for animals. Emotional clarity showed a significantly stronger correlation with the human empathy sub-dimensions of perspective-taking ($Z = 2.46$, $p < 0.05$), fantasy ($Z = 2.14$, $p < 0.05$), and personal distress ($Z = 3.46$, $p < 0.01$) than with empathy for animals. Finally,

**Table 4 Summary of the hierarchical multiple regression analysis.**

| Step | Sample | Criterion | Predictors | B | SE | Beta | t | p |
|---|---|---|---|---|---|---|---|---|
| 1 | 471 | Empathy for animals | Pet owner | −0.10 | 2.51 | −0.18 | −4.13 | <0.001 |
| | | | Gender | 0.12 | 2.61 | 0.19 | 4.43 | <0.001 |
| | | | Age | −0.31 | 0.04 | −0.11 | −2.50 | <0.05 |
| 2 | 471 | Empathy for animals | Pet owner | −10.95 | 2.50 | −0.19 | −4.38 | <0.001 |
| | | | Gender | 10.65 | 2.64 | 0.18 | 4.03 | <0.001 |
| | | | Age | −0.24 | 0.13 | −0.08 | −1.85 | 0.66 |
| | | | Attention (TMMS) | 3.60 | 1.53 | 0.11 | 2.33 | <0.05 |
| | | | Clarity (TMMS) | −2.71 | 1.67 | −0.08 | −1.63 | 0.10 |
| | | | Repair (TMMS) | 4.40 | 1.74 | 0.12 | 2.52 | <0.05 |
| 3 | 471 | Empathy for animals | Pet owner | −10.96 | 2.46 | −0.19 | −4.50 | <0.001 |
| | | | Gender | 5.92 | 2.78 | 0.10 | 2.13 | <0.05 |
| | | | Age | −0.26 | 0.13 | −0.09 | −2.00 | <0.05 |
| | | | Attention (TMMS) | 0.95 | 1.70 | 0.03 | 0.57 | 0.57 |
| | | | Clarity (TMMS) | −2.17 | 1.70 | −0.06 | −1.30 | 0.20 |
| | | | Repair (TMMS) | 2.92 | 1.78 | 0.08 | 1.64 | 0.10 |
| | | | Perspective-taking (IRI) | −0.31 | 2.04 | −0.00 | −0.15 | 0.87 |
| | | | Fantasy (IRI) | 3.13 | 2.01 | 0.08 | 1.55 | 0.12 |
| | | | Empathic concern (IRI) | 9.97 | 2.50 | 0.22 | 4.07 | <0.001 |
| | | | Personal distress (IRI) | −1.52 | 1.80 | −0.04 | −0.85 | 0.34 |

emotional repair showed a significantly stronger relationship with the human empathy sub-dimensions of fantasy ($Z = 2.67$, $p < 0.01$) and personal distress ($Z = 4.82$, $p < 0.01$) than with empathy for animals. Finally, we did not find significant differences between the animal empathy and EI correlation when the sample was divided into pet owners and non-pet owners ($Z = 0.76$, NS, for attention to emotions, $Z = 0.22$, NS, for emotional clarity NS, $Z = 1.29$, NS, for emotional repair).

In order to examine the study variables that can best predict empathy for animals, a hierarchical regression analysis was carried out with empathy for animals as criterion, and pet ownership, gender and age (Step 1), sub-dimensions of EI (Step 2), and sub-dimensions of human empathy (Step 3) as predictors. The significant predictors of empathy for animals included in the first step were pet ownership, gender and age; in the second step these were pet ownership, gender, and the attention and repair sub-dimensions of EI, and in the third step these were pet ownership, age, gender and the empathic concern sub-dimension of human empathy (see Table 4 for details). The final model accounted for 16.1% of the variance in empathy for animals.

Finally, moderation analyses were conducted with EI and the human empathy sub-dimensions that were significant predictors of empathy for animals. The results revealed that having a pet had a moderating effect on the relationship between empathic concern and empathy for animals, indicating that this relationship was stronger in participants who had pets (see Fig. 1; interaction effect = −10.41, 95% CI [−18.38, −2.44], ($p < .05$)). The rest of the sub-dimensions did not show a significant moderation effect.

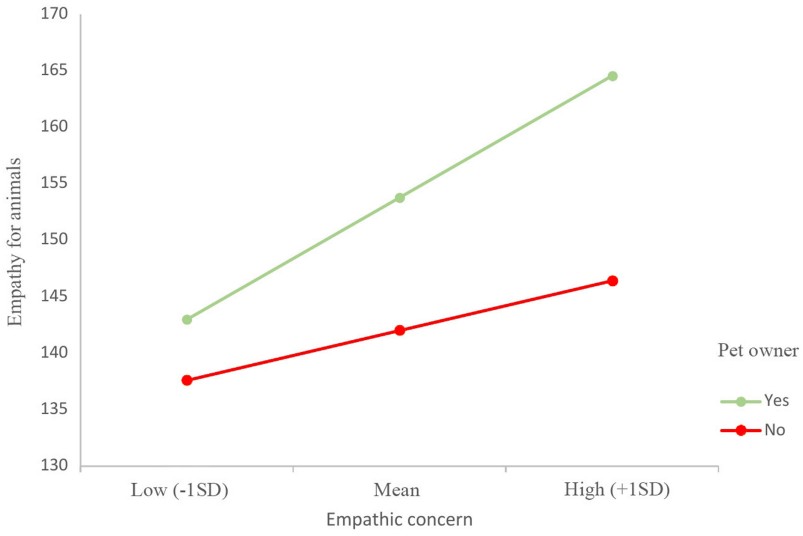

**Figure 1 Effect of empathic concern on empathy for animals, moderated by pet ownership.** Values on the *y*-axis represent the levels of empathy for animals. Values on the *x*-axis represent levels of empathic concern where high and low were specified at −1 SD (low) and +1 SD (high) of the centered means.

## DISCUSSION

Previous studies in the literature have shown that EI is related to positive aspects, such as better mental health, greater prosocial behaviours, and greater human empathy (*Ciarrochi, Chan & Caputi, 2000*; *Fitness & Curtis, 2005*; *Martins, Ramalho & Morin, 2010*). However, to date, the relationship between EI and empathy for animals has not been studied, despite the fact that animals are an increasingly important part of our society and everyday living. The present study attempted to delve more deeply into the relationship between EI and empathy for humans and for animals. More in-depth knowledge about these factors could help us to understand the differences that exist between personal emotional capacities and sensitivity to animals.

First, our study revealed that women, compared with men, showed a higher score on the EI sub-dimension of attention to emotions. This result is consistent with the previous literature and supports the hypothesis that women appear to have a greater ability to recognise feelings and know their meaning (*Cabello & Fernández-Berrocal, 2015*; *Fernández-Berrocal & Extremera, 2008*). We also found that women, compared with men, obtained higher scores on all the human empathy sub-dimensions. Empirical studies have indicated that women have a greater capacity than men for understanding the thoughts and feelings of others (*Klein & Hodges, 2001*; *Schieman & Van Gundy, 2000*). Finally, we observed that women scored significantly higher than men on empathy for animals. This finding is in accord with various studies showing that women tend to show a more positive attitudes towards animals (*Furnham, McManus & Scott, 2003*; *Paul, 2000*; *Serpell, 2004*). With regard to differences according to pet ownership, we found that pet owners showed higher scores on empathy for animals than those participants who did not

have a pet. These results are in accord with those reported in the previous literature, showing that familiarity with animals increases empathy for them (*Paul, 2000*).

With respect to the Hypotheses 1 and 2, our results were consistent with most of the findings in the literature (*Ellingsen et al., 2010*; *Extremera & Fernández-Berrocal, 2004*; *Findlay, Girardi & Coplan, 2006*; *Fitness & Curtis, 2005*; *Junttila et al., 2006*). Regarding Hypothesis 1, the results confirmed that, in general, an adequate level of EI was related to higher levels of human empathy (*Fitness & Curtis, 2005*; *Mayer, Caruso & Salovey, 1999*; *Schutte et al., 2001*). However, it must be noted that an excess of empathic involvement (i.e., higher scores on personal distress) could hinder the ability to engage in emotionally intelligent behaviours (*Extremera & Fernández-Berrocal, 2004*). This latter assumption could explain the observation that levels of personal distress were negatively related to emotional clarity and repair. With regard to Hypothesis 2, we found a positive relationship between most of the human empathy sub-dimensions and empathy for animals. Specifically, this study found that the emotional aspect of human empathy (the sub-dimension of empathic concern) was the best predictor of empathy for animals. This finding is also in accord with previous research (*Ellingsen et al., 2010*; *Emauz et al., 2016*; *Paul, 2000*) and suggests that those individuals with higher scores on human empathy also have a more welfare-oriented attitude towards animals. Whilst several theories could explain this result, in general, research indicates that someone who is empathetic, that is, capable of adopting the point of view of animals, and exhibits concern about them is likely to have similar feelings towards people (*Eisenberg et al., 1992*; *Lockwood, 1983*; *Messent, 1983*; *Rossbach & Wilson, 1992*).

To address Hypothesis 3, we analyzed the relationship between EI and empathy for animals. Analysis of the total sample revealed that higher levels of empathy for animals were only positively related to the sub-dimension of attention to emotions. However, when we introduced EI, together with human empathy, as a predictor of empathy for animals, only the relationship with the empathic concern sub-dimension remained significant.

Finally, in order to verify if these relationships depended on the degree of familiarity (in terms of pet ownership) that the person has with animals (Hypothesis 4), we conducted further analyses by dividing the participants into two samples, that is, pet owners and non-pet owners. Two main results were found. First, for the sample of pet owners, the results revealed a positive correlation between the sub-dimensions of attention to emotions and repair with empathy for animals, whilst analysis of the sample of non-pet owners did not yield any significant relationship between EI and empathy for animals. Second, the relationship between empathic concern and empathy for animals was significantly stronger for those participants who had pets compared with those without pets. These findings suggest that interaction with animals can influence the relationship between EI, human empathy, and empathy for animals. Moreover, these results could also reflect a potential bidirectional effect of the association established between empathy for animals and pet ownership; individuals who have/had a pet report greater empathy for animals, but also people with greater empathy for animals are also more likely to have pets and enjoy or value their company.

In summary, the current findings do not support the notion that people who have better emotional abilities are more empathetic towards animals. Our results instead appear to support the idea that the proficiency for understanding and managing emotions is developed on the basis of human interactions and such emotional abilities are insufficient to perceive and understand the emotional signs of animals and consequently empathise with them. Direct interaction with animals would thus be needed to improve these emotional abilities. To the best of our knowledge, this is the first time that the relationship between EI and empathy for animals has been investigated. Although EI has been linked to better interpersonal social relationships, prosocial behaviour, and greater empathy for humans (*Brackett, Mayer & Warner, 2004*; *Gilet et al., 2013*; *Komorosky & O'Neal, 2015*; *Lopes et al., 2011*), the results of our study indicate that EI may not be a determining factor in empathy for animals, or at least suggests that the mechanisms underlying both types of empathy are influenced by different factors (*Paul, 2000*, *2005*).

The present study provides a first step towards a better understanding of the relationships between empathy for humans and animals. However, it is important to consider that the methodology employed was correlational and thus future lines of investigation should conduct experimental studies to determine causality between variables. Advances in the study of this relationship could have practical implications such as the promotion of interventions aimed at increasing human empathy levels through animal-assisted therapy. This type of therapy could be helpful for decreasing antisocial behaviours and aggressiveness among peers and, in addition, could promote appropriate attitudes and respect for animal welfare.

As limitations of the research, it is important to note that our sample was not gender matched, with a greater number of women than men (34.4% were men). Moreover, given that previous literature has shown that the ability to empathize is influenced and reinforced by similarity, age, gender, factors related to theory of mind or personality traits (*De Waal, 2008*; *Kavanagh, Signal & Taylor, 2013*), future studies should explore possible differences in the empathy and EI relationship as a function of these factors. It is also important to note that, due to the correlational-transversal design of this research, further exploration of the potential bidirectionality effects among the variables could not be tested in this study (e.g., empathy for animals and pet ownership).

Future research studies should aim to replicate these results in other countries, since there are cultural differences that could modify people's empathic values with respect to animals (*Young, Khalil & Wharton, 2018*). In addition, the degree of proximity to animals has only been operationalized in terms of pet ownership, thus excluding people who have other types of close relationships with animals, such as farmers, or people that work in other similar professions. Therefore, future research should explore the various types of relationships that humans can have with animals. Finally, the questionnaires used in this research were self-report instruments, the responses of which are susceptible to possible response and introspective biases. In a similar vein, the reliability of the IRI questionnaire was not very high (Cronbach's alpha ranged between 0.66 and 0.76). In future investigations it might therefore be useful to work with behavioural and performance measures in order to address these issues.

## CONCLUSION

The main objective of the current research was to clarify the relationship between EI and empathy for humans and animals in order to have a better understanding of the factors that underlie empathic behaviour towards animals and its relationship with empathy for humans. Our results revealed the existence of a positive relationship between both types of empathy (humans and animals). A positive relationship between EI and empathy for humans was also observed. However, these relationships were dependent on the participants' experience with animals. Overall, these results support previous literature regarding the positive relationship between EI and empathy for humans, but the mixed findings observed between EI and empathy for animals suggest a greater complexity in the relationships between these constructs, perhaps indicating that both types of empathy can be guided by different factors or represent different psychological constructs. Although preliminary conclusions can be drawn from these results, further investigation is necessary in order to replicate these findings and better understand the common and distinctive process involved in empathy for humans and animals, and their association with EI abilities.

### Funding

This work was funded by the Spanish Ministry of Economy, Industry and Competitiveness (project: PSI2017-84170-R) to Pablo Fernández-Berrocal, Junta de Andalucía (project: UMA18-FEDERJA-137) to Alberto Megías-Robles, the Ministry of Education and Vocational Training (FPU15/05179) to Raquel Gómez-Leal, and by the Portuguese National Science Foundation (Postdoctoral Grant awarded to Ana Costa and supervised by Luísa Faria - FCT SFRH/BPD/117479/2016). The funders had no role in study design, data collection and analysis, decision to publish, or preparation of the manuscript.

### Grant Disclosures

The following grant information was disclosed by the authors:
Spanish Ministry of Economy, Industry and Competitiveness: PSI2017-84170-R.
Junta de Andalucía: UMA18-FEDERJA-137.
Ministry of Education and Vocational Training: FPU15/05179.
Portuguese National Science Foundation: FCT SFRH/BPD/117479/2016.

### Competing Interests

The authors declare that they have no competing interests.

### Author Contributions

- Raquel Gómez-Leal conceived and designed the experiments, performed the experiments, prepared figures and/or tables, authored or reviewed drafts of the paper, and approved the final draft.

- Ana Costa conceived and designed the experiments, analyzed the data, prepared figures and/or tables, authored or reviewed drafts of the paper, and approved the final draft.
- Alberto Megías-Robles conceived and designed the experiments, performed the experiments, authored or reviewed drafts of the paper, and approved the final draft.
- Pablo Fernández-Berrocal conceived and designed the experiments, analyzed the data, authored or reviewed drafts of the paper, and approved the final draft.
- Luísa Faria analyzed the data, authored or reviewed drafts of the paper, and approved the final draft.

### Human Ethics

The following information was supplied relating to ethical approvals (i.e., approving body and any reference numbers):

The Research Ethics Committee of the University of Málaga approved the study protocol as part of the project PSI2017-84170-R (IRB approval number CEUMA 14-2019-H).

### Data Availability

Complete data for the variables included in the study are available in the Supplemental File.

### Supplemental Information

Supplemental information for this article can be found online at http://dx.doi.org/10.7717/peerj.11274#supplemental-information.

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
