# Peer review of "Relationship between emotional intelligence and empathy towards humans and animals"

_PeerJ, doi:10.7717/peerj.11274_

## Round 0.1 · original submission · Major Revisions

The reviewers returned disparate verdicts on the publishability of your MS in its current form but both agree that the paper is interesting and well written. However, both also identified some key shortcomings with your approach.

Reviewer 1 points out an important omission in that we do not know anything about the participants’ experience with animals. I agree that this should be addressed.

Reviewer 2 notes that you have not defined empathy toward animals and this is also an extremely critical omission. You did not provide any sample items for the AES.

Both of these omissions will have to be remedied in a revision.
This is a relatively small sample for this type of work and there is no reason not to collect additional data – both to see if the findings are replicable in another sample and to address the shortcoming identified by Reviewer 1. I would also be inclined to include a measure of theory of mind.

I also read this paper with interest as I have conducted some related work myself regarding empathy for humans and animals (as well as examining the link between emotional intelligence, theory of mind, and empathy). I think the association with emotional intelligence and empathy for animals could be very interesting. However, I think this part of the rationale for your study is underdeveloped. You need to lay more of the groundwork for how and why EI and empathy are related and then provide more of an analysis as to the relevance for empathy for animals. Because EI is based upon an individual’s understanding of their own emotion states, and reading animal emotions is clearly quite distinct from the complexities and nuances involved in human interactions, one might easily expect the lack of an association between EI and empathy for animals, even though it exists for empathy toward humans. I liked your discussion of why empathy for humans and animals might not always be strongly related and I think this discussion could be better connected to the role of EI.

Why did you not also analyze the data using regression to get at each component’s unique associations with the empathy outcomes?

The lines 97-100 don’t make a lot of sense. Isolated cases don’t indicate that there is a lack of data on the relations; but they may point to greater complexity in the relations than one might intuit.

Reviewer 1 ·

Basic reporting

No comment.

Experimental design

No comment.

Validity of the findings

The authors provide meaningful replication of previous findings on the relationship between human empathy and animal empathy and emotional intelligence and empathy. That said, I do have a concern that the authors do not control for current or childhood pet ownership within their analyses. Paul (2000) mentions that animal empathy was significantly different between owners and non-owners, which could fundamentally change the findings if it is controlled for in the current analyses. Without that control, the paper may contribute inaccurate findings to the literature. I should note that I'm not concerned with the current lack of findings between EI and animal empathy (and if they still lack significance when controlling for ownership, that would also be noteworthy), but that the findings could be wrong. At minimum there should be a discussion of this point in the limitation section, as it is a primary finding of the seminal work on animal empathy and a paper that is cited.

Additional comments

Overall an interesting paper but I would strongly encourage reporting if past/current ownership was assessed. If so, test it. If not, be sure to mention it as a limitation of the study to provide a clear path for future research.

·

Basic reporting

Thank you for the opportunity to review this interesting paper. It is well-written and the Introduction provides a very sound review of the literature on EI and empathy. The presentation of the article, including its tables, is professional; and the raw data and ethics statements have been shared. There is a major problem with the paper, however, that makes it difficult to evaluate in regard to its contribution to the scientific literature. Specifically, I needed much more information about the construct of 'empathy for animals' before I could assess the logic of the hypotheses. What does the construct mean, and how has it previously been measured? Is it about compassion for animals and if so, all of them or just a few species? Is it about cognitive empathy, involving respect for animals and a moral imperative not to cause them unnecessary harm? Is it about feeling a close, empathic relationship with animals that means one tends to treat them more like humans? These are just speculations, because no information about the scale used to measure the construct was provided in the Method section, making it impossible to understand the nature of the associations reported in the Results. My suggestion for improvement (because I think the topic is interesting) is for the authors to re-conceptualize their study with a very clear focus on the construct of empathy for animals - make this the central feature of the study and describe in detail how the construct is measured. This will make the interpretation and discussion of the Results much more meaningful.

Experimental design

No further comment

Validity of the findings

No further comment

Additional comments

I commend the authors on putting together such a well-written, professionally presented paper and encourage them to submit a new paper with a strong conceptual and methodological focus on empathy for animals and its implications.

---

## Round 0.2 · Minor Revisions

The paper now is better structured and there is a clearer rationale for the study. It reads well and will eventually make a nice contribution in an understudied area. It is great that you have been able to add data on pet ownership. However, you have not followed the advice for revising your analytic strategy and I would encourage you to do so.

It strikes me that the finding is a bit circular. People who have experience with pets may report greater empathy for animals b/c people with greater empathy for animals are more likely to have pets and otherwise engage with them. Please consider in your discussion.
On line 99, I think you could be more specific to say that empathy for animals may not always relate to empathy for humans rather than a vague platitude about complexity.

Is there a citation for the point you make on line 205 about displacing affection from humans to animals, which is very interesting?

Power analyses should be conducted prior to performing the study to avoid over or undersampling. Using it as a post-hoc measure to determine adequacy after the fact is not appropriate. Now you are left needing to justify why you oversampled. I would delete lines 274—277. My previous point was not that you did not have enough power to detect effects but that it would be prudent to see if the effects replicated in another sample.

Your reliability for the IRI is a bit low. Do you have concerns about the Spanish version? I think this should be addressed at least in the discussion. How do you expect your Spanish participant data to relate to data from other countries?

As Reviewer 2 points out, you have not followed the advice to use a regression approach to your data. This is in fact the preferred approach for correlational data. Correlations and regressions provide different pieces of information about the association between your variables and although it is appropriate to present correlational matrices, a regression analysis should also be presented. Your approach does not allow you to examine moderation or mediation. The suggested approach would also allow you to determine if there are possibly suppression effects etc. It seems that your hypothesis might be that the association between empathy for humans and empathy for animals is mediated by EI and moderated by experience with animals/pets and gender. Thus you should develop and analyze such a model.

Reviewer 1 ·

Basic reporting

I appreciate the author(s) efforts to strengthen their work in response to my comments. That said, I think displaying the findings by ownership may be beneficial for readers to better understand those findings, esp. since that seems to be a major finding (i.e., that direct interaction is needed to empathize with animals).

Experimental design

no comment

Validity of the findings

no comment

·

Basic reporting

Thank you for the opportunity to review this revised version of the paper, and please accept my apologies for lateness. Overall, I believe the authors have done a very good job in revising their paper in line with previous comments and recommendations. The coverage of the relevant literature is both wider and deeper, and the rationale for the study is more compelling than previously. In particular, the authors have clarified their construct of 'empathy for animals' and I appreciated the addition of scale items to demonstrate how the construct was measured. The quality of English used throughout the paper is good, with just a few words that are a little out of place: e.g., line 154, "objects" should probably be "entities"; and pg. 188, 'stimulated' should probably be 'simulated'. Line 194 also needs attention: specifically, an individual observing an animal in a state of suffering MAY (not will) experience distress.." This may not apply to a hunter, for example, who may experience happiness at shooting an animal. Finally, the first paragraph of the Results also needs some attention - it's always good to see effect sizes reported but this section is a bit hard to follow.

Experimental design

The Methodology is fine and. as noted above, it was good to see some example items from the Animal Empathy Scale. The sample size is a particular strength. My only remaining caveat concerns the rationale, expressed just before the Methods section, for operationalizing '"degree of proximity that the person has with animals" in terms of pet ownership only. My thoughts throughout the paper go to farmers and others who have working relationships - sometimes very close - with animals that are not pets (including race horses; experimental animals in laboratories such as mice or even chimps), and that may be hurt or even slaughtered at some point. Many farmers claim to feel enormous empathy with their stock and do everything in their power to give them "happy" lives; others regard them as units of production and have no emotions towards them at all. I think the "pet" relationship is too narrow and doesn't tap into the wide range of potential relationships that people may have with animals. I realize that this can't be remedied in the current paper, but I believe it is a limitation worth mentioning in the Discussion, especially in the context of future research.

Validity of the findings

One point: in response to the editor's query about why the author(s) did not use a (more economical) regression analysis, they claim that they didn't do this because they were not predicting causality between the constructs analysed. However, regression is a correlational model and does not imply causality - the use of the word "predictor" in regression models often trips people up! I still think it would be worthwhile - and cleaner - to take a regression approach.

Additional comments

Overall, a great topic that I hope you will pursue further.

---

## Round 0.3 · accepted · Accept

Thank you for adding the additional analysis and making the final edits. During proofs, please add the p value on line 406 for the interaction effect. I am happy to see this paper published as it represents an understudied research area.